

# Phytochemical-rich medicinal plant extracts suppress bacterial antigens-induced inflammation in human tonsil epithelial cells

Niluni M. Wijesundara[1], Satvir Sekhon-Loodu[1] and HP Vasantha Rupasinghe[1,2]

[1] Department of Plant, Food, and Environmental Sciences, Faculty of Agriculture, Dalhousie University, Truro, NS, Canada
[2] Department of Pathology, Faculty of Medicine, Dalhousie University, Halifax, NS, Canada

## ABSTRACT

**Background**. Pharyngitis is an inflammatory condition of the pharynx and associated structures commonly caused by the Group A streptococci (GAS). There is a growing interest in discovering plant-based anti-inflammatory compounds as potential alternatives to conventional drugs. This study evaluated anti-inflammatory activity of phytochemical-rich extracts prepared from 12 herbal plants using human tonsil epithelial cells (HTonEpiC) *in vitro*.

**Methods**. The HTonEpiC were induced by a mixture of lipoteichoic acid (LTA) and peptidoglycan (PGN) (10 µg/mL; bacterial antigens) for 4 h and then exposed to ethanol extracts (EE) or aqueous extracts (AE) for 20 h. The secretion of four pro-inflammatory cytokines was measured using enzyme-linked immunosorbent assays (ELISA). Total phenolic and total flavonoid contents of the extracts were determined using spectrophotometric methods.

**Results**. The herbal plant extracts ($\leq$5 µg/mL) were not cytotoxic to HTonEpiC. The extracts exhibited a broad range of reduction (1.2%–92.6%) of secretion of interleukin-8 (IL-8), human beta defensin-2 (hBD-2), epithelial-derived neutrophil activating protein-78 (ENA-78), and granulocyte chemotactic protein-2 (GCP-2). Both EE and AE of clove, ginger, and echinacea flower and EE from danshen root significantly inhibited the pro-inflammatory cytokine production as induced by LTA and PGN in HTonEpiCs at the concentrations of 1 and 5 µg/mL.

**Discussion**. Our observations indicate that danshen root, clove, ginger, and echinacea flower extracts exhibit an anti-inflammatory effect in HTonEpiCs. The most efficacious extracts from danshen root, clove, ginger and echinacea flowers have potential to be used as natural sources for developing phytotherapeutic products in the management of painful inflammation due to streptococcal pharyngitis.

Corresponding author
HP Vasantha Rupasinghe,
vrupasinghe@dal.ca

# INTRODUCTION

The prevalence of streptococcal pharyngitis has increased worldwide during the last decades. *Streptococcus pyogenes*, a group A streptococcus (GAS), is the main bacterial etiology
responsible for 15%–36% of acute pharyngitis in children (*Abachi, Lee & Rupasinghe, 2016*). The inflammatory response of epithelial cells of the upper respiratory tract act as the first line of defense recruited to combat GAS (*Rock et al., 2010*). Toll-like receptors (TLR) in host cells are involved in the recognition of cell wall compounds of *S. pyogenes,* especially lipoteichoic acid (LTA) and peptidoglycan (PGN), which are responsible for host-pathogen interactions (*Bisno, Brito & Collins, 2003*). Bacterial virulence factors induce tonsil epithelial cells and white blood cells to produce chemical mediators of inflammation such as cytokines, chemokines, and prostaglandins (*Ricciotti & FitzGerald, 2011*).

Non-steroidal anti-inflammatory drugs (NSAID), such as aspirin, ibuprofen, nimesulide, piroxicam, and ketoprofen are widely recommended by physicians for the management of inflammatory conditions (*Cremonesi & Cavalieri, 2015*). NSAID inhibit the cyclooxygenase-2 (COX-2), which is responsible for the synthesis of pro-inflammatory prostaglandins (*Ricciotti & FitzGerald, 2011*). Some patients are allergic to NSAID and may develop shortness of breath after intake (*Kim et al., 2013*). Physicians reluctant to give steroids to children due to possible long-term side effects (*Schams & Goldman, 2012*).

Pharmaceutical and natural health product industries have been interested in identifying specific medicinal plants as sources of unique phytochemicals with pharmacological properties. *Oregano* L., *Salvia* L. and *Thymus* L., three important genera of the Lamiaceae family, are spice herbs that are traditionally used for flavoring food in North America (*Fournomiti et al., 2015*). Spices such as ginger and clove are also used globally in food flavoring, confectioneries, beverages, and cosmeceuticals (*Chaieb et al., 2007*; *Park, Bae & Lee, 2008*). Native Americans and early settlers used Echinacea, geranium, slippery elm, barberry and licorice as medicinal herbs. Traditionally, the tea made from the leaves of Echinacea and geranium or roots of barberry and licorice have been largely used in North America as a soothing agent for infections and inflammations (*Borchers et al., 2000*).

Thus, the aim of this study was to investigate the anti-inflammatory properties of aqueous and ethanol extracts prepared from 12 herbs, which were selected based on the reported literature and geographical availability and traditional practice in Canada. Cell viability, production of pro-inflammatory biomarkers, and cell morphological changes were determined using human tonsil epithelial cell (HTonEpiC) model to identify efficacious extracts, with the future aim of developing an herbal extract incorporated natural health product in the management of streptococcal pharyngitis associated inflammation.

## MATERIALS AND METHODS

### Plant materials

Twelve different herbal plants that were used in Canadian traditional medicine were selected for the study. Purple coneflower/Echinacea areal parts (Voucher No: 13009), geranium leaves (Voucher No: 13010), sage leaves (Voucher No: 13011), oregano flowering shoots (Voucher No: 13012), and thyme flowering shoots (Voucher No: 13013), were collected during the flowering period from the herb garden at the Faculty of Agriculture, Dalhousie University (GPS location 45°22′23.3″N 63°15′45.2″W). A taxonomist, Jeff Morton, Faculty of Agriculture, Dalhousie University, Canada, authenticated plants. The specimens were

deposited in the A.E. Roland herbarium, Department of Plant, Food, and Environmental Sciences, Faculty of Agriculture, Dalhousie University, Canada. Fresh ginger rhizome, and dried clove flower buds were purchased from Halifax (NS, Canada). A dry powder of barberry root, licorice root, slippery elm/red elm inner bark and olive leaves were purchased from Mother Earth Natural Health, Ottawa, ON, Canada. Danshen roots were obtained from Green Man Botanicals (Gaspereau Mountain, NS, Canada). Samples were dried at 50 °C for 3 days, ground, and stored at −80 °C.

## Preparation of extracts of medicinal plants
### Aqueous extracts (AE)
The AEs were prepared using a previously described method (*Gunathilake & Rupasinghe, 2014*). Plant powder and distilled water (1:10) was boiled using a ISOTEMP™ water bath (Model 205; Fisher Scientific, Ottawa, ON, Canada) for 10 min and were filtered. Collected filtrates were freeze overnight and were dried in a freeze dryer (Kinetics; FTS Systems Inc., Stone Ridge, NY, USA) under 3,600 mT vacuum and −20 °C for 48 h and were stored in airtight amber glass bottles at −80 °C.

### Ultrasonic-assisted ethanol extracts (EE)
The mixture of plant powder and 95% ethanol (1:10) kept in a sonication bath (750D; VWR, West Chester, PA, USA) at 35 °C for 45 min (3 × 15 min with 5 min of intervals) at 40 kHz frequency and 150 W ultrasonic power. The residues were removed by filtering through a vacuum pump and were evaporated to dryness using a rotary evaporator (R-200; Buchi, Flawil, Switzerland) at 45 °C for 20–30 min. Remained solids were dissolved in anhydrous ethanol and kept under the nitrogen evaporator (N-EVAP™; Organomation Association Inc., Berlin, NJ, USA). After completely dry, the extracts were preserved in airtight amber glass bottles at −80 °C.

## Phytochemical analysis
### Determination of total phenolic (TP) content
The Folin Ciocalteu assay was performed to estimate the TP content using a previously described method (*Rupasinghe et al., 2010*). The TP was calculated using a standard curve prepared with gallic acid and were expressed as milligrams of gallic acid equivalents (GAE) per gram of dry solid of extract.

### Determination of total carotenoids (TC) content
The TC assay was performed using a previously described method (*Rivera & Canela, 2012*). The TC values were calculated using the equation; $TC = (A \times 10{,}000 \times V)/(A(1\%/1 \text{ cm}) \times W \times L)$, where $A$ is the absorbance at 470 nm, $V$ is the total volume of extract, $A^{(1\%/1 \text{ cm})}$ is the extinction coefficient for a mixture of solvents arbitrarily set at 2,500, $W$ is the sample weight in grams and $L$ is the path length for the sample volume in the plate. TC content was expressed as µg/mL of dry solid of extract.

## Cell culture
The HTonEpiC cells (ScienCell Research Laboratory, San Diego, CA, USA) were cultured and maintained according to the manufacturer's guidelines. Briefly, HTonEpiC were

cultured in poly-L-lysine (PLL) (Sigma-Aldrich Canada Ltd., Oakville, ON, Canada) coated flask (2 $\mu g/cm^2$ T-75 flask) with complete growth medium (CGM) and maintained at 37 °C in a 5% $CO_2$ humidified atmosphere in an incubator (Model 3074; VWR International, West Chester, PA, USA). CGM was prepared by mixing growth supplement and penicillin/streptomycin solution with tonsil epithelial cell medium at 1:1:100 ratio.

## Cell viability assay

Cell viability was determined using the 3-(4,5-dimethylthiazol-2-yl)-5-(3-carboxymethoxyphenyl)-2-(4-sulfophenyl)-2H-tetrazolium (MTS) (Sigma-Aldrich Canada Ltd., Oakville, ON, Canada) assay. Briefly, cells were cultured at a density of 6,000 cells/100 $\mu L$ in 96-well plates pre-coated with PLL and treated with EE and AE at concentrations of 0.5–100 $\mu g/mL$, solvent control (0.05% DMSO) and Dulbecco's phosphate buffered saline buffer. A combination of lipoteichoic acid (LTA) (Sigma-Aldrich Canada Ltd., Oakville, ON, Canada) and peptidoglycan (PGN) (Cedarlane Laboratories, Burlington, ON, Canada) (10 $\mu g/mL$) was used as a bacterial antigen control and nimesulide (0.5, 1, and 5 $\mu g/mL$) was tested as a positive control. After 24 h incubation under 5% $CO_2$ at 37 °C, the cells were refreshed by adding 100 $\mu L$ of fresh CGM. Then cells were incubated with 20 $\mu L$ of MTS reagent (MTS + phenazine methosulfate (PMS)) for 2.5 h. The absorbance was measured at 490 nm and cell viability was calculated using the following equation.

$$\text{Cell Viability (\%)} = \frac{\text{Absorbance of the treated wells} - \text{Absorbance of the blank}}{\text{Absorbance of the control wells} - \text{Absorbance of the blank}} \times 100,$$

where the treated wells contained cells incubated with test compounds, the control wells contained cells with solvent and CGM, and the blank wells contained CGM only.

## Treatment of plant extracts

The HTonEpiC were seeded at a density of $35 \times 10^3$ cells/well in PLL coated 24-well plates and incubated overnight at 35 °C for 20 h. The cells were stimulated with a mixture of LTA and PGN (each at 10 $\mu g/mL$) for 4 h incubation at 35 °C to trigger inflammation. Then, the cells were treated with either EE or AE at 1 and 5 $\mu g/mL$ concentrations and incubated at 35 °C for 24 h. The cells were stimulated with LTA and PGN for 4 h and 24 h. Nimesulide as the reference control and 0.05% of DMSO as the experimental control were used. The supernatants were collected for determination of pro-inflammatory biomarkers.

## Determination of pro-inflammatory cytokines and chemokines

The concentrations of pro-inflammatory cytokines were measured from the culture medium of control and treated cells by using ELISA kits according to the manufacturer's instructions.

### Interleukin-8 (IL-8) assay

The concentration of IL-8 was measured by using the IL-8 ELISA kit (BD Biosciences, Mississauga, ON, Canada). Anti-human monoclonal antibodies coated plates were developed by using detection antibodies and streptavidin-horseradish peroxidase (HRP) conjugate provided with the kit, according to the instructions. First, 50 $\mu L$ diluent was added to each well of anti-human IL-8 monoclonal antibody coated 96-well plates, and then standards and

samples (100 μL) were pipetted into appropriate wells. Plates were incubated for 2 h at room temperature with gentle shaking. Subsequently, well-washing step, 100 μL of detecting antibody was added. The covered plates were incubated for 1 h at room temperature, followed by seven rinsing steps. Then, 100 μL/well of 3, 3′, 5, 5′-tetramethylbenzidine (TMB) substrate was added and were incubated for 30 min at room temperature in the dark. Then, the stop solution (0.16 M sulfuric acid) was added (50 μL/well) and absorbance was read at 450 nm. The concentration of IL-8 was expressed as pg/mL using the standard curve and presented as the percentage of the inflammation control.

### Human Beta Defensin-2 (hBD-2) assay

The content of hBD-2 was determined by an hBD-2 ELISA kit (PromoCell GmbH; Sickingenstrabe, Heidelberg, Germany). Anti-hBD-2 antibody coated plates were prepared using detection antibody and avidin-HRP conjugate, according to the manufacturer's protocol. The 96-well plates were coated with capture antibody and incubated overnight at room temperature. Samples/standards (100 μL) were added in triplicates to appropriate wells after rinsing. Followed by a 2 h incubation at room temperature, plates were washed four times with washing buffer. Then, the plates were incubated for another 2 h at room temperature along with detection antibody. The avidin-HRP conjugate was pipetted and incubated for 30 min. 2, 2′-Azino-bis (3-ethylbenzothiazoline-6-sulfonic acid) liquid substrate was added for color development and were read at the absorbance of 405 nm.

### Epithelial-derived Neutrophil Activating protein-78 (ENA-78) assay

The human ENA-78 ELISA kit (Ray Biotech, Inc., Norcross, GA, USA) was used to detect the % secretion of the ENA-78 protein in the cell culture supernatant. Pre-coated 96-well plates with specific antibody for human ENA-78 were used and the assay was performed according to the manufacturer's instructions. Briefly, 100 μL standards and samples were added to appropriate wells and the plates were incubated overnight at 4C with gentle shaking. The solution was then discarded and the plates were washed four times with wash solution. Then, 100 μL of biotinylated antibody was added to each well and incubated for 1 h at room temperature with gentle shaking. After four washing steps, 100 μL of HRP-streptavidin solution was added to each well. After an incubation period of 45 min at room temperature with gentle shaking, the plates were rinsed with wash buffer, followed by 100 μL of TMB substrate reagent was added to each well and incubated for 30 min at room temperature with gentle shaking, protected from light. The reaction was ended by adding 50 μL of stop solution (0.16 M sulfuric acid). Then, plates were read at the absorbance of 450 nm. The ENA-78 concentration was calculated using a standard curve and the data is expressed as pg/mL.

### Granulocyte Chemotactic Protein-2 (GCP-2) assay

The human GCP-2 ELISA kit (Ray Biotech; Inc., Norcross, GA, USA) was used to measure the protein production of GCP-2. GCP-2 microplate coated with anti-human GCP-2 was used. The ELISA procedure was similar to the description in the ENA-78 assay.
## Cell morphological assessment

After the treatment with 1 and 5 μg/mL of extracts, the cells were examined under an inverted microscope (ECLIPSE TS 100/TS 100-F; Nikon Instruments Inc., Melville, NY, USA) with 40 × magnification. The images were captured and saved using a Lumenara infinity camera (1–2 USB, 2.9 Megapixel), coupled with capture and analyzing software (Infinity Analyze; Lumenara Corporation, Ottawa, ON, Canada).

## Statistical analysis

All the experiments were designed using completely randomized design. Cell viability experiments were conducted in triplicate and independently three times whereas all the ELISA experiments were run in triplicate and independently twice. Results were expressed as a mean ± standard error of the mean. One-way ANOVA analysis was performed by using Minitab 17.0 statistical software and statistical differences ($P < 0.05$) between means of pairs were resolved by using Tukey's tests.

# RESULTS

## Extraction yield and phytochemical analysis of herbal extracts

Phytochemical analysis of selected herbal plant parts used in this study have been reported excessively in past. Table 1 summarizes the scientific names, major phytochemicals and their therapeutic uses of the plants used in the present study. The extraction yields of AE, EE, TP, and TC contents are shown in Table 2. Other than, in licorice EE, the extraction yields of AEs were higher than their respective EEs. However, no exact relationship was observed between the plant parts have been used in the extraction and the extraction yield. Both EE and AE of clove displayed the highest TP content, followed by AE and EE of oregano, slippery elm, thyme, geranium, and ginger. The EE from leaves were showed higher TC content than EEs derived from roots, flowers or stem. Therefore, thyme, oregano, sage, echinacea leaves and geranium exhibited significant high TC. Furthermore, EEs showed higher TC content than their respective AEs.

## Cytotoxic effect of selected phytochemicals-rich extracts on human tonsil epithelial cells

The suppressive effect of extract is expected only to happen on pro-inflammatory cytokines production in inflamed cells. If normal throat epithelium cells are affected by treated extracts, they may not use as anti-inflammatory agents in natural health product or in drugs. Therefore, cytotoxicity of both EEs and AEs on HTonEpiC were evaluated by MTS assay. The HTonEpiC were incubated 24 h with 14 EE and 13 AE at various concentrations ranged 0.5–100 μg/mL. The geranium leaves showed cytotoxic effects at ≥1 μg/mL for EE and at >5 μg/mL for AE (Table S1). The EE of sage leaves were cytotoxic at higher concentrations than 5 μg/mL, where % cell viabilities were 33.1 ± 1.7, 29.5 ± 0.8, 24.4 ± 0.9 and 25.3 ± 1.8 for sage EE and at the concentrations of 10, 25, 50, and 100 μg/mL, respectively. Similarly, 25, 50, and 100 μg/mL concentrations of EE of licorice were also showed cytotoxicity to HTonEpiC. All other extracts showed significantly higher cell viability (>80%) without demonstrating cytotoxicity to the HTonEpiC when the concentrations were between 0.5

**Table 1** Major phytochemicals reported for the selected herbal plant parts used in this study.

| Plant name | Family | Parts used | Major phytochemicals | References |
|---|---|---|---|---|
| Barberry (*Berberis vulgaris* L.) | Berberidaceae | R | Berberine, Berbamine, 5-Methoxyhydnocarpin, Berlambine, Jatrorrhizine, Palmatine, Quercetin, Rutin and Oxyberberine | *Abd El-Wahab et al. (2013)* and *Mokhber-Dezfuli et al. (2014)* |
| Clove (*Syzygium aromaticum* L.) | Myrtaceae | FB | Eugenol, Eugenyl acetate, β-Caryophyllene, α-Humulene, β-Ocimene, Caryophyllene oxide, α-Copaene and p-Allyl phenol | *Chaieb et al. (2007)* and *Rani et al. (2012)* |
| Echinacea/purple cone flower (*Echinacea purpurea* L.) | Asteraceae | L, S, F | Caftaric acid, Chlorogenic acid, Caffeic acid, Cynarin, Echinacoside, Cichoric acid, Quercetin and Kaempferol | *Dennehy (2001)* and *Sharma et al. (2010)* |
| Geranium (*Pelargonium graveolens* L.) | Geraniaceae | L | Citronellol, Citronellyl formate and Geraniol | *Ghannadi et al. (2012)* |
| Ginger (*Zingiber officinale* L.) | Zingiberaceae | Rh | Gingerol, Galanolactone, Ginerdiol, Geranial, α-Zingiberene, Oleoresin, and Zingerone | *Gunathilake & Rupasinghe (2014)* and *Young et al. (2005)* |
| Licorice (*Glycyrrhiza glabra* L.) | Fabaceae | R | Glycyrrhizin, Glabridin, Liquirtin, Licoflavan, Narigenin and Asparegene | *Chu et al. (2012)* and *Shin et al. (2008)* |
| Olive (*Olea europeus* L.) | Oleaceae | L | Hydroxytyrosol, Luteolin-7-O-glucoside, Luteolin-4-O-glucoside, Oleuropein, and Hydroxytyrosol acetate | *Goulas et al. (2009)* |
| Oregano (*Origanum vulgare* L.) | Lamiaceae | FS | Carvacrol, p-Cymene, Borneol, Thymol, Linalool, Linlyl acetate, Terpinene-4-ol, Rosmarinic acid, Thymohydroquinone and Naringin | *Fournomiti et al. (2015)* and *Teixeira et al. (2013)* |
| Sage (*Salvia officinalis* L.) | Lamiaceae | L | 1,8-Cineole, p-Cymene, Camphor, borneol, α-Thujone, β-Pinene, α-Humulene, trans-Caryophyllene, β-Thujone and Myrcene | *Fournomiti et al. (2015)*, *Poeckel et al. (2008)* |
| Danshen (*Salvia miltiorrhiza* Bunge.) | Lamiaceae | R | danshensu, protocatechuic aldehyde, salvianolic acid B, cryptotanshinone, tanshinone I and tanshinone IIa | *Chang et al. (2008)* |
| Slippery elm (*Ulmus rubra* Muhl) | Ulmaceae | IB | Oleanolic acid, Ursolic acid, Uvaol, Betulinic acid, Botulin, β-Arotene, β-Sitosterol and Citrostadienol | *Lesley (2006)* |
| Thyme (*Thymus vulgaris* L.) | Lamiaceae | FS | Thymol, γ-Terpinene, p-Cymene, Mycrene, α-Pinene, α-Thujone, α-Terpinene, Carvacrol, 1,8 Cineole, Methyl ether, Linalool, γ-Terpinene and α-Terpineol | *Fachini-Queiroz et al. (2012)* and *Fournomiti et al. (2015)* |

Notes.

F, flowers; FB, flowering buds; IB, inner bark; FS, flowering shoots; Rh, rhizome; R, roots; L, leaves; S, stem.

**Table 2 Extraction yield, total phenolic content, and total carotenoids content of the examined plant extracts.**

| Plant name | Parts used | Extract yield (%) (DW basis) | | Total phenolics (mg GAE/g DW) | | Total carotenoids (μg/g DW) | |
|---|---|---|---|---|---|---|---|
| | | EE | AE | EE | AE | EE | AE |
| Barberry | R | $1.9 \pm 0.0^g$ | $5.9 \pm 0.1^e$ | $2.6 \pm 0.05^f$ | $2.3 \pm 0.1^e$ | $494.9 \pm 4.0^e$ | $31.6 \pm .7^e$ |
| Clove | FB | $21.3 \pm 1.2^b$ | $24.6 \pm 0.4^{bcd}$ | $19.1 \pm 0.0^a$ | $16.0 \pm 0.2^a$ | $92.1 \pm 0.2^{hi}$ | NA |
| Echinacea | L | $7.1 \pm 0.3^{efg}$ | $31.5 \pm 1.7^a$ | $1.7 \pm 0.0^{fg}$ | $2.1 \pm 0.1^e$ | $6248.8 \pm 11.6^a$ | $24.3 \pm 0.9^{ef}$ |
| | S | $5.6 \pm 0.0^{fg}$ | $18.9 \pm 0.4^{cd}$ | $1.1 \pm 0.3^g$ | $1.2 \pm 0.1^f$ | $388.0 \pm 3.7^f$ | $147.0 \pm 2.1^d$ |
| | F | $6.8 \pm 0.1^{efg}$ | $20.3 \pm 1.3^d$ | $1.0 \pm 0.0^g$ | $2.6 \pm 0.1^e$ | $292.6 \pm 0.6^g$ | $2.0 \pm 0.0^h$ |
| Geranium | L | $9.3 \pm 0.1^{def}$ | $29.0 \pm 2.2^{ab}$ | $7.1 \pm 0.4^{cd}$ | $10.6 \pm 0.3^b$ | $2505.8 \pm 8.4^b$ | $18.6 \pm 3.4^{fg}$ |
| Ginger | Rh | $19.6 \pm 0.9^{bc}$ | $19.9 \pm 0.7^d$ | $6.9 \pm 0.5^{cd}$ | $0.5 \pm 0.4^f$ | $455.0 \pm 19.7^{ef}$ | $7.8 \pm 0.2^{gh}$ |
| Licorice | R | $39.0 \pm 1.9^a$ | $19.5 \pm 0.2^d$ | $4.6 \pm 0.0^e$ | $2.8 \pm 0.1^e$ | $157.4 \pm 3.9^h$ | $237.3 \pm 0.4^c$ |
| Olive | L | $22.2 \pm 2.7^{bc}$ | $24.2 \pm 2.3^{bcd}$ | $5.3 \pm 0.0^e$ | $5.7 \pm 0.2^c$ | $235.6 \pm 2.3^g$ | $677.6 \pm 4.2^a$ |
| Oregano | FS | $14.5 \pm 0.5^{cd}$ | $8.2 \pm 0.2^e$ | $9.2 \pm 0.5^b$ | $9.8 \pm 0.1^b$ | $2032.4 \pm 24.7^d$ | $10.6 \pm 0.3^{gh}$ |
| Sage | L | $8.9 \pm 0.9^{def}$ | $19.8 \pm 1.1^d$ | $5.2 \pm 0.1^e$ | $4.1 \pm 0.1^d$ | $2066.9 \pm 11.8^d$ | $25.9 \pm 1.0^{ef}$ |
| Danshen | R | NA | NA | $10.0 \pm 0.5^b$ | NA | NA | NA |
| Slippery elm | IB | $7.6 \pm 0.3^{ef}$ | $7.1 \pm 0.1^e$ | $7.5 \pm 0.1^c$ | $1.9 \pm 0.1^e$ | $72.7 \pm 6.2^i$ | $400.5 \pm 3.5^b$ |
| Thyme | FS | $10.5 \pm 0.4^{de}$ | $18.6 \pm 1.1^d$ | $6.0 \pm 0.2^{de}$ | $5.1 \pm 0.1^c$ | $2373.2 \pm 27.7^c$ | $401.5 \pm 4.9^b$ |

**Notes.**

Means $\pm$ SEM ($n = 3$) of different extracts analyzed individually in triplicate. Different superscript letters within the same column indicate significant differences of means among extraction solvents used in the study.

AE, aqueous extracts; DW, dry weight of extracts; EE, ethanol extracts; GAE, Gallic acid equivalents; NA, not analyzed; TE, Toluene equivalents; F, flowers; FB, flowering buds; IB, inner bark; FS, flowering shoots; Rh, rhizome; R, roots; L, leaves; S, stem.

and 5 μg/mL in AE and cell viability % were shown in Table 3 (Concentrations for 10, 25, 50, and 100 μg/mL were tested; Tables S1 and S2). Bacterial antigen mixture (PGN + LTA) had no significant cytotoxicity at the tested concentration of 10 μg/mL ($96.4 \pm 1.6\%$ of cell viability for AE and $96.8 \pm 1.1\%$ for EE). Nimesulide showed low cytotoxicity to the cells at all the concentrations tested. The concentrations of 1 and 5 μg/mL of extracts were chosen to induce cells for inflammation study.

## Effect of phytochemical-rich extracts on morphological changes of human tonsil epithelial cells

Further verification of morphological alterations was attained from phase contrast microscopy. We examined the HTonEpiCs that were treated with CGM, 0.05% DMSO, 10 μg/mL LTA + PGN and plant extracts at the concentration of 5 μg/mL (Figs. 1 and 2). The 5 μg/mL extracts showed significantly higher suppression of pro-inflammatory cytokines than 1 μg/mL extracts ($p \leq 0.05$). Therefore, only the results of % total production of biomarkers of LTA and PGN-induced HTonEpiC at 5 μg/mL are reported. No changes in cell morphology and density were observed between untreated and DMSO-treated HTonEpiCs. Some morphological changes are shown after treatment of AEs (Fig. 1) and EEs (Fig. 2) at the concentrations of 5 μg/mL such as changes in size, shape, and reduction of cell density due to inflammation. Moreover, treatment with 10 μg/mL LTA + PGN, promoted the most significant damages to the cell morphology and density reduction compared to untreated and herbal extract treated HTonEpiCs.

**Table 3  Effect of concentration of the extracts on percentage viability of human tonsil epithelial cells.**

| Test materials | Cell viability (%) | | | | | |
|---|---|---|---|---|---|---|
| | 0.5 µg/mL | | 1 µg/mL | | 5 µg/mL | |
| | AE | EE | AE | EE | AE | EE |
| Licorice R | 96.4 ± 0.9[a] | 99.1 ± 0.9[a] | 97.1 ± 0.5[a] | 98.0 ± 0.9[a] | 80.7 ± 1.0[c] | 98.4 ± 0.9[a] |
| Sage L | 98.7 ± 0.6[a] | 99.8 ± 0.2[a] | 98.7 ± 0.5[a] | 96.2 ± 3.4[a] | 89.4 ± 2.9[b] | 86.2 ± 0.6[c] |
| Echinacea S | 97.8 ± 0.6[a] | 82.8 ± 2.1[c] | 98.0 ± 0.5[a] | 95.8 ± 2.1[a] | 84.2 ± 2.0[c] | 83.0 ± 1.2[c] |
| Echinacea F | 98.2 ± 0.3[a] | 97.1 ± 1.2[a] | 98.1 ± 0.0[a] | 96.1 ± 1.7[a] | 99.1 ± 1.8[a] | 97.1 ± 0.6[a] |
| Oregano FS | 99.3 ± 1.1[a] | 96.9 ± 1.1[a] | 97.1 ± 0.0[a] | 96.9 ± 0.6[a] | 97.4 ± 0.6[a] | 86.9 ± 0.6.[c] |
| Thyme FS | 99.1 ± 0.6[a] | 95.6 ± 1.8[ab] | 96.1 ± 0.6[a] | 96.6 ± 1.8[ab] | 98.9 ± 0.5[a] | 95.6 ± 1.2[a] |
| Barberry R | 98.1 ± 0.6[a] | 97.5 ± 0.4[a] | 97.0 ± 0.7[a] | 95.5 ± 0.2[ab] | 96.0 ± 0.3[a] | 97.5 ± 0.6[a] |
| Slippery elm IB | 99.0 ± 0.8[a] | 93.4 ± 0.6[b] | 97.0 ± 0.6[a] | 89.7 ± 0.6[b] | 97.6 ± 1.1[a] | 82.7 ± 0.6[c] |
| Clove FB | 98.4 ± 0.0[a] | 97.5 ± 1.1[a] | 98.9 ± 0.2[a] | 96.5 ± 1.1[ab] | 98.3 ± 0.6[a] | 97.5 ± 0.7[a] |
| Ginger Rh | 98.1 ± 0.0[a] | 98.9 ± 0.3[a] | 98.4 ± 0.5[a] | 98.9 ± 0.9[a] | 98.7 ± 0.3[a] | 98.9 ± 0.7[a] |
| Olive L | 97.0 ± 0.9[a] | 96.8 ± 0.9[ab] | 96.0 ± 0.9[a] | 98.8 ± 0.9[a] | 96.2 ± 0.6[a] | 98.8 ± 1.2[a] |
| Geranium L | 98.2 ± 0.7[a] | 85.1 ± 1.1[c] | 99.0 ± 0.6[a] | 46.1 ± 1.1[de] | 93.1 ± 0.6[ab] | 41.1 ± 0.0[e] |
| Echinacea L | 98.0 ± 0.8[a] | 97.3 ± 0.8[a] | 96.3 ± 0.9[a] | 97.9 ± 0.9[a] | 97.1 ± 0.75[a] | 97.9 ± 0.3[a] |
| Danshen R | ND | 95.9 ± 0.6[ab] | ND | 98.9 ± 0.6[a] | ND | 98.9 ± 0.6[a] |
| Nimesulide | 98.0 ± 0.9[a] | 97.4 ± 0.5[a] | 98.1 ± 0.8[a] | 95.8 ± 1.0[ab] | 96.0 ± 0.6[a] | 95.8 ± 1.0[ab] |

Notes.
Cells were treated with various concentrations of the test materials for 24 h. Cell viability (%) was calculated relative to the control of 0.05% DMSO. Values of the same column are expressed as mean ± SEM ($n = 3$).
[a–g]The value with different letters indicates the significant difference determined by Tukey's test ($p \leq 0.05$).
ND, not determined; F, flowers; FB, flower bud; FS, flowering shoots; IB, inner bark; L, leaves; Rh, rhizome; R, roots; S, stem.

## Inhibitory effects of selected phytochemical-rich extract on LTA-PGN stimulated secretion of pro-inflammatory biomarkers

In this study, the levels of pro-inflammatory cytokines and chemokines released into the CGM were measured by ELISA. Bacterial antigen-stimulated HTonEpiC cell model was employed. HTonEpiC were incubated with a mixture of LTA and PGN (10 µg/mL) to induce inflammation. Our results showed that secretion of IL-8, hBD-2, ENA-78, and CGP-2, by with the presence of all of the tested extracts, except geranium EE, were significantly declined in a dose-dependent manner. The 5 µg/mL extracts showed significantly higher suppression of pro-inflammatory cytokines than 1 µg/mL extracts ($p < 0.05$). In particular, treatment of cells with 5 µg/mL plant extracts reduced the secretion of pro-inflammatory markers tested in a range of 1.2%–92.6% (Figs. 3 and 4). Both the extracts of ethanol and aqueous geranium showed significant reduction of cytokine, but in spite of anti-inflammatory potential, it was excluded because of its cytotoxicity, which might have been responsible for the decreased levels of IL-8, hBD-2, ENA-78, and GCP-2 (Figs. 3 and 4).

## DISCUSSION

Phytochemicals have been recognized to possess biological activities against upper respiratory infections and associated inflammation (*Hostanska et al., 2011*; *Khouya et al., 2015*; *Sharma et al., 2010*). The present study was conducted to assess the anti-inflammatory

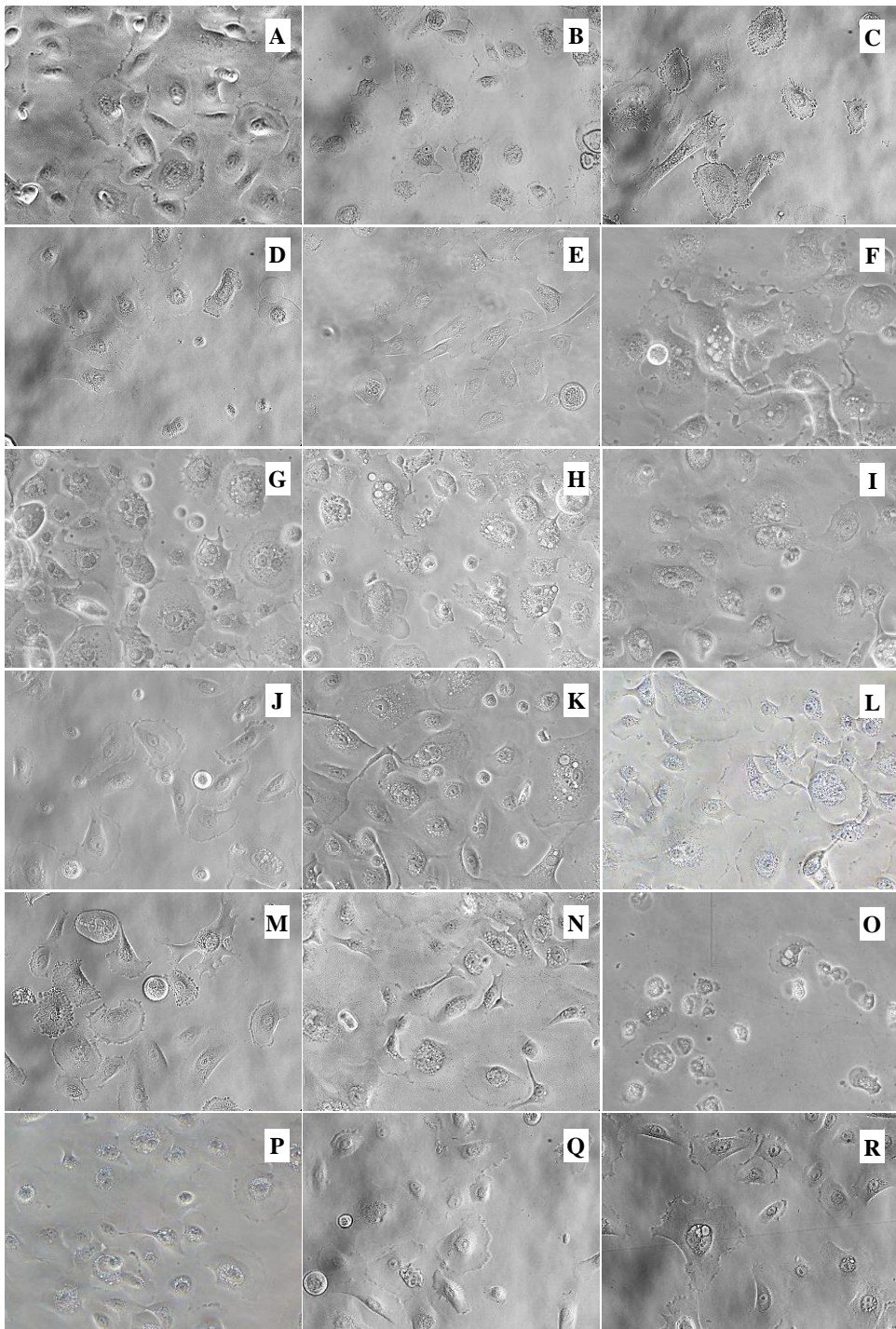

**Figure 1 Morphology of inflammation-induced tonsil epithelial cells treated with the ethanol extracts.** The LTA + PGN-induced cells were incubated with 5 μg/mL of the extracts for 24 h. All images were obtained at a magnification of ×40. 

**Figure 1 (. . . continued)**
(A) Media control; (B) 0.05% DMSO control; (C) 10 μg/mL LTA and PGN; (D) Licorice root; (E) Sage leaves; (F) Echinacea stem; (G) Echinacea flower; (H) Oregano shoots; (I) Thyme shoots; (J) Barberry root; (K) Slippery elm inner bark; (L) Clove flower bud; (M) Ginger rhizome; (N) Olive leaves; (O) Geranium leaves; (P) Echinacea leaves; (Q) Danshen root and (R) Nimesulide.

properties, the secretion of pro-inflammatory biomarkers by bacterial antigen-induced human epithelial cells.

In streptococcus pharyngitis, host cells first recognize cell surface components of *S. pyogenes*, such as LTA and PGN and then respond to GAS through the innate immune system (*Vroling, Fokkens & Drunen, 2008*). The TLR play a key role in regulating the expression of pro-inflammatory mediators that are involved in the inflammatory and immunological reactions (*White, 1999*). Macrophages and mast cells induce the release of inflammatory mediators, such as COX-2, *i*NOS, and inflammatory cytokines. Inflammatory responses have long been considered associate with the nuclear factor-kappa B (NFκB) signaling pathway (*Ghosh, May & Kopp 1998*) that involved in the induction of the expression of pro-inflammatory genes, including many cytokines, chemokines, and other adhesion molecules, in the inflammatory response and the adhesion of *S. pyogenes* alone might be sufficient to stimulate nuclear translocation of NF-κB (*Medina, Anders & Chhatwal, 2002*).

The percentage suppression of IL-8 by danshen root EE (46.7%), ginger EE (46.1%), echinacea flower EE (45.8%), and slippery elm EE (44.3%) was significant ($P < 0.05$) at the concentration of 5 μg/mL, when compared to 5 μg/mL of nimesulide (54.3%) and 1 μg/mL of nimesulide (52.9%). Also, clove AE (62.1%), echinacea flower AE (59.8%), oregano flowering shoots AE (56.9%), echinacea leaves AE (56.5%) and sage leaves AE (54.8%) showed significantly higher suppression than the reference drug. These findings are in agreement with other studies that showed diminished expression of various cytokines, including IL-8 by herbal extracts (*Hostanska et al., 2011*; *Sharma et al., 2010*). For example, "echinaforce", a compound found in EE of herb and roots of *E. purpurea* L. (echinacea), inhibited several bacterial inductions of various cytokines, including IL-4, IL-6, IL-8, TNF-$\alpha$ and monocyte chemo-attractant protein-1, in a human tracheobronchial epithelial cell line (BEAS-2B) and a human lung epithelial cell line (A-549) (*Sharma et al., 2010*). Possible reasons for this lower activity of danshen root EE, ginger EE, echinacea flower EE and slippery elm EE may be because of the chemical constituents such as flavonoids, quinones, alkaloids, triterpenes, and polyacetylates.

In the present study, clove, ginger, echinacea flower, oregano, thyme, and sage as well as clove, ginger, and danshen EE, which were effective inhibitors of chemokine IL-8 production, also potently inhibited hBD-2 production. A previous study has shown that the levels of hBD-2 strongly correlated with those of increased IL-8 synthesis by lung epithelial cells *in vitro* (A549) and in human primary bronchial epithelial cell lines (*Van Wetering et al., 1997*). An herbal formulation (BNO 1030), an extract of seven herbal drugs, showed anti-inflammatory properties at low non-cytotoxic concentrations by suppressing the secretion of IL-8 and hBD-2 in cultured epithelial A549 cells (*Hostanska et al., 2011*).

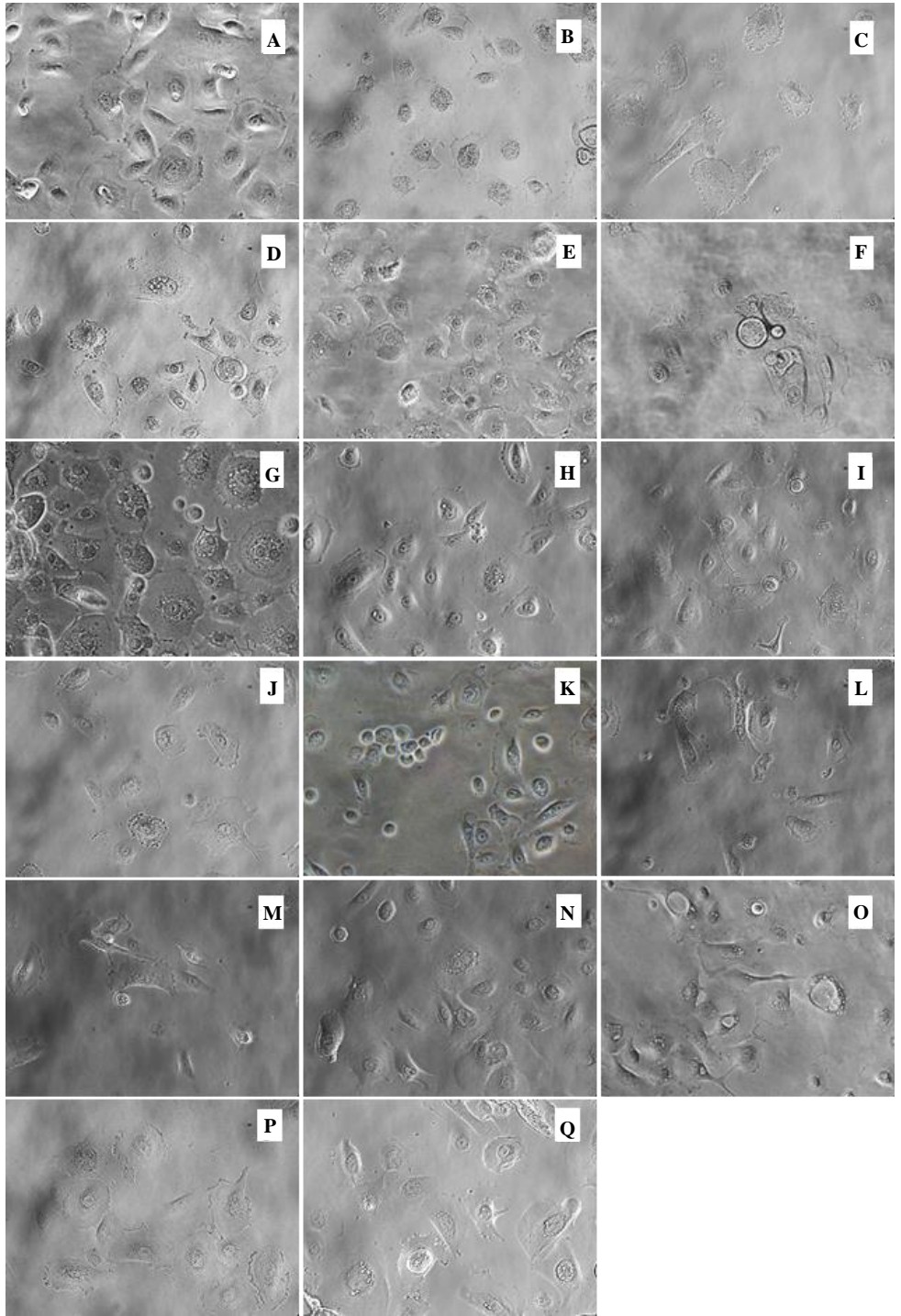

**Figure 2 Morphology of inflammation-induced tonsil epithelial cells treated with the aqueous extracts.** The LTA + PGN-induced cells were incubated with 5 μg/mL of the extracts for 24 h. All images were obtained at a magnification of ×40. (continued on next page...)

**Figure 2 (…continued)**
(A) Media control; (B) 0.05% DMSO control; (C) 10 μg/mL LTA and PGN; (D) Licorice root; (E) Sage leaves; (F) Echinacea stem; (G) Echinacea flower; (H) Oregano shoots; (I) Thyme shoots; (J) Barberry root; (K) Slippery elm inner bark; (L) Clove flower bud; (M) Ginger rhizome; (N) Olive leaves; (O) Geranium leaves; (P) Echinacea leaves and (Q) Nimesulide.

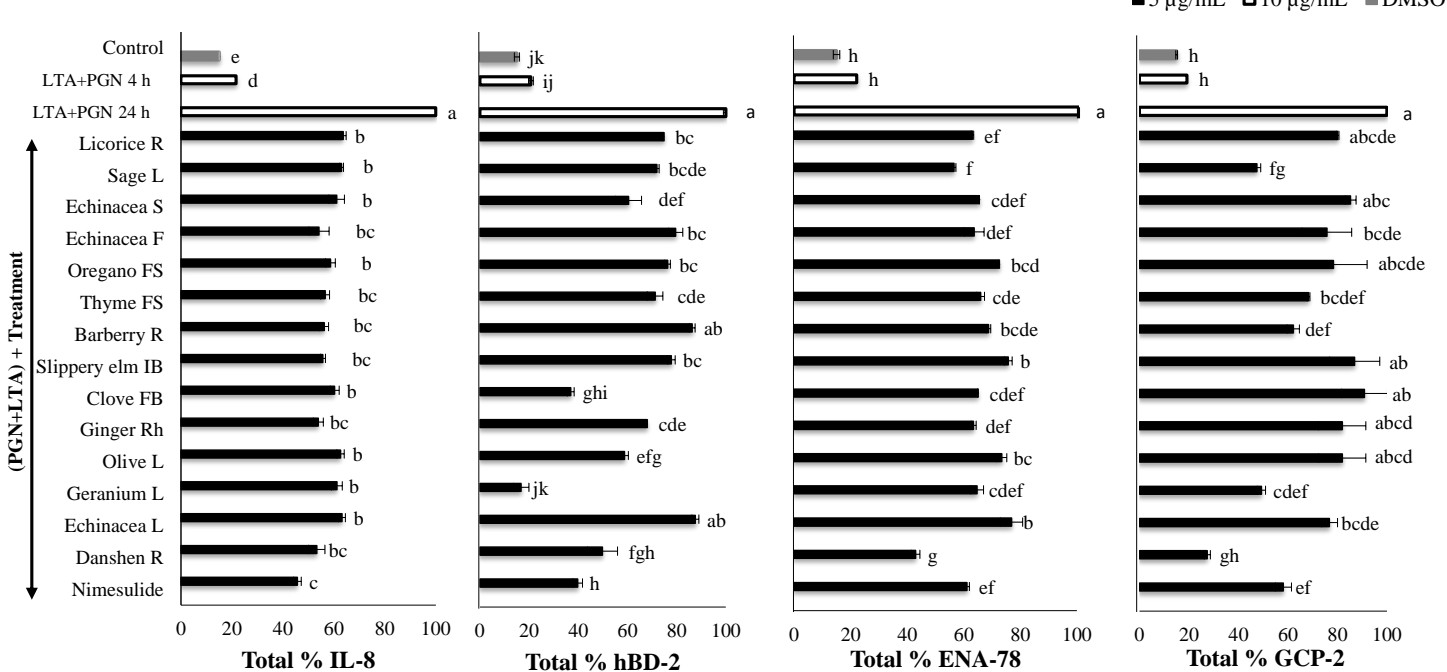

**Figure 3** **Secretion of pro-inflammatory proteins by tonsil epithelial cells treated with the ethanol extracts.** The inflammation of human tonsil epithelial cells was stimulated by a mixture of LTA + PGN, and then incubated with selected phytochemical-rich ethanol extracts. Data were expressed as mean ± SEM ($n = 3$). Groups sharing different letters showed a significant difference (Tukey's test, $p \leq 0.05$). IL-8, interleukin-8; hBD-2, human beta defensin-2; GCP-2, granulocyte chemotactic protein-2; ENA-78, epithelial-derived neutrophil activating protein-78; LTA, lipoteichoic acid; PGN, peptidoglycan; F, flowers; FB, flower bud; FS, flowering shoots; IB, inner bark; L, leaves; Rh, rhizome; R, roots; S, stem.

It also explained that phenolic compounds, such as flavonoids, tannins and phenolic acids present in BNO 1030, possessed cytokine suppressive capacity. The significant suppression of the production of hBD-2 and IL-8 by AE of thyme, oregano, echinacea, sage, clove, and ginger as well as EEs of danshen root, ginger, and clove may also have explained due to their phenolic compounds. Interestingly, the abovementioned extracts showed significantly higher suppression of hBD-2 than nimesulide, suggesting potential clinical applications. However, this correlation between hBD-2 and IL-8 is biologically relevant because both hBD-2 and IL-8 inhibit cytokine are produced by T helper 2 cells (*Nomura et al., 2003*). This finding agreed with the synergistic support of hBD-2 and IL-8 expression for the T helper 1 cells, as described previously (*Meyer et al., 2006*). The activation of NF-κB and AP-1 is a compulsory prerequisite for increased IL-8 and IL-6 expression in epithelial cells, in response to *S. pyogenes* infection and inflammation responses.

Examining the % expression of GCP-2, it was found that the EE of danshen root, clove, ginger, thyme, oregano, sage, licorice, echinacea flower, and stem, as well as all the AE,

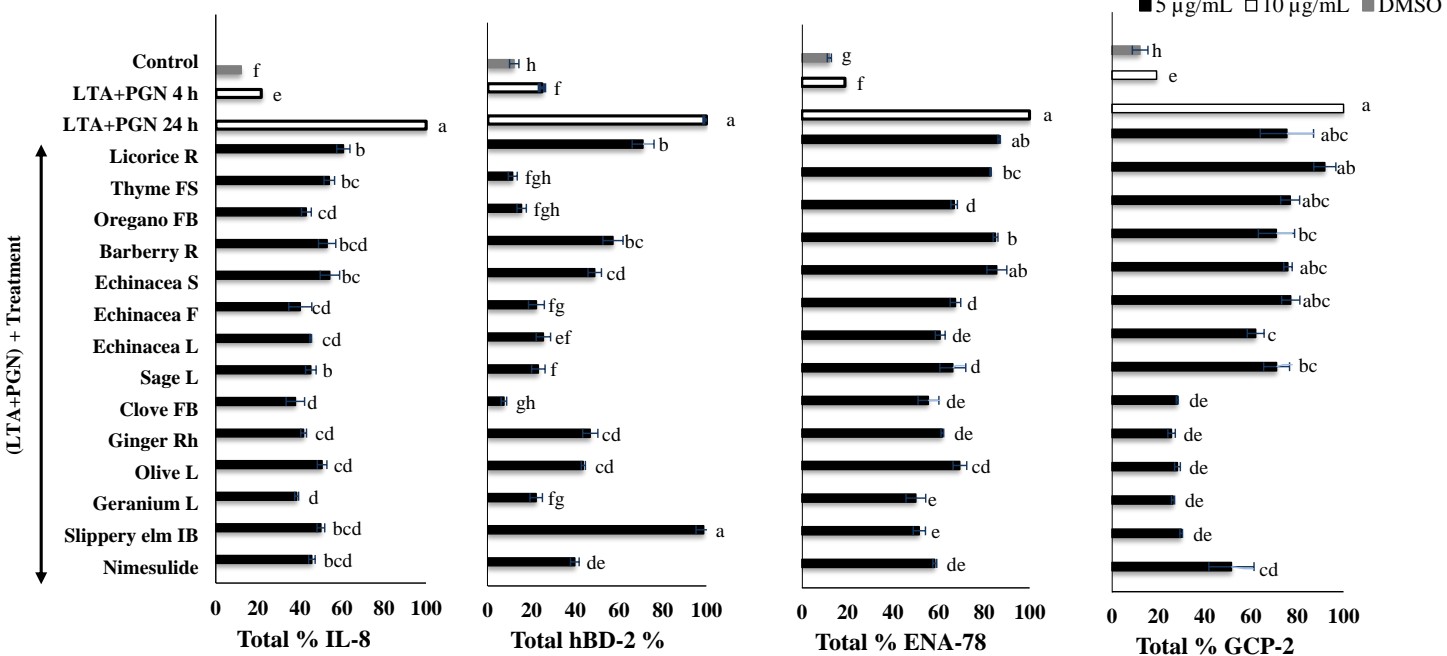

**Figure 4** Secretion of pro-inflammatory proteins by tonsil epithelial cells treated with the aqueous extracts. The inflammation of human tonsil epithelial cells was stimulated by a mixture of LTA + PGN, and then incubated with selected phytochemical-rich ethanol extracts. Data were expressed as mean ± SEM ($n = 3$). Groups sharing different letters showed a significant difference (Tukey's test, $p \leq 0.05$). IL-8, interleukin-8; hBD-2, human beta defensin-2; GCP-2, granulocyte chemotactic protein-2; ENA-78, epithelial-derived neutrophil activating protein-78; LTA, lipoteichoic acid; PGN, peptidoglycan; F, flowers; FB, flower bud; FS, flowering shoots; IB, inner bark; L, leaves; Rh, rhizome; R, roots; S, stem.

account for the significant reduction of the secretion of a pro-inflammatory biomarker by LTA- and PGN-induced HTonEpiC (Figs. 3 and 4). Furthermore, suppression of ENA-78 (59.7%) when treated with 5 µg/mL sage root EE was shown to be significantly higher than that of 5 µg/mL nimesulide (41.6%). Nevertheless, it should be noted that there was a significant reduction of ENA-78 by 5 µg/mL EE of clove, sage leaves, licorice root, ginger, echinacea flower and thyme, as 37.7%, 46.2%, 39.7%, 39.3%, 39.0% and 36.7%, respectively. Among AE, clove, echinacea leaves, and ginger showed the highest suppression ($P < 0.05$) of LTA and PGN on ENA-78 secretion. Although GCP-2 is structurally closely related to ENA-78, functionally it can be used by both GCP-2 and IL-8 receptors to chemo-attract neutrophils (*Mittal et al., 2014*; *Wuyts et al., 2003*). Therefore, GCP-2 was shown to have higher antibacterial activity against *S. pyogenes,* compared with ENA-78 and other chemokines. The angiogenic properties of GCP-2, along with its chemotactic property, cause this enhanced antibacterial activity (*Sachse et al., 2005*). There was an abundant expression of IL-8 and GCP-2 in the surface epithelium of an acute type of tonsillitis and ENA-78 was almost undetectable (*Sachse et al., 2005*). This is contrary to the present results, where compared to GCP-2, the pronounced ENA-78 protein was expressed by phytochemical-rich extracts. A possible reason may be the difference in the cell models used by the two studies.

Conventional therapy, steroidal and NSAID are used to treat acute inflammation. However, some of those drugs exhibit several side effects (*Kim et al., 2013*). Therefore,

alternative treatments, with safer compounds, have still to be discovered. Combined treatment of thyme and oregano essential oils, has been shown to limit the production of pro-inflammatory cytokines, and reduce 2,4,6-trinitrobenzene sulphonic acid-induced colitis in mice (*Bukovska et al., 2007*). The thyme aromatic oil contained about 48% *p*-cymene and 24% of thymol, while oregano aromatic oil, which contained about 55% of carvacrol, was the major active compound of these extracts (*Bukovska et al., 2007*). Moreover, *in vivo* anti-inflammatory activities of EE of water pepper (*Polygonum hydropiper* L.) (*Yang et al., 2012*) and EE of Chinese cinnamon (*C. cassia* L.) (*Yu et al., 2012*) suppressed the production of nitric oxide, TNF-$\alpha$, and PG E2, in LPS-activated RAW264.7 cells, along with peritoneal macrophages, in a dose-dependent manner.

Among the tested herbs, EE of danshen root, ginger, clove, echinacea flower and AE of clove, ginger, echinacea flower significantly ($p \leq 0.05$) diminished the LTA- and PGN-induced pro-inflammatory cytokines secretion. Danshen root extract was the most potent inhibitor of production of IL-8, hBD-2, ENA-78 and GCP-2. These results were consistent with the previous reports that demonstrate there are active anti-inflammatory phytochemicals of different plant extracts. For example, previous studies have reported that analgesic and anti-inflammatory activities of various phytochemicals: suppression of the transcription of pro-inflammatory mediators by [6]-gingerol of ginger (*Young et al., 2005*); anti-inflammatory activities in $H_2O_2$-stimulated macrophages by *E. angustifolia* L. extract (*Pomari, Stefanon & Colitti, 2014*); and inhibition of *i*NOS expression by carnosic acid and carnosol present in sage (*S. officinalis* L.) extracts (*Poeckel et al., 2008*).

Although both EE and AE from licorice root and barberry suppressed the cytokines production in the present study, the percentage reductions of some cytokines were significantly lower than that of ginger, clove, and echinacea extracts. However, anti-inflammatory effects of licorice root extracts and their phytochemicals such as glycerrhitinic acid, glycyrrhizin, licochalcone, and glycerol have been reported (*Chu et al., 2012*; *Shin et al., 2008*). The anti-inflammatory activities of the major compounds present in the clove, ginger, sage as well as echinacea extracts were reported earlier and this evidence elucidates the present results (Table 1). The present study also found that echinacea, sage, and oregano contain a higher content of total phenolic and carotenoids. The efficacious extracts or isolated phytochemicals from selected plant sources can be used as an ingredient for anti-inflammatory natural health product manufacturing for soothing sore throat and tonsillitis in streptococcus pharyngitis.

## CONCLUSION

This study demonstrated that anti-inflammatory properties of danshen root EE, clove AE and EE, ginger AE and EE, echinacea flower AE and EE, oregano leaves AE, sage leaves EE or thyme flowering shoot AE as evident by their ability to suppress protein production of pro-inflammatory mediators such as IL-8, hBD-2, CGP-2 and ENA-78 *in vitro*. These efficacious extracts have the potential to be used in developing natural health products such as phytochemicals incorporated lozenges or herbal teas designed for managing pharyngitis with the aim of relieving the complications associated with inflammatory conditions.

Further studies are warranted to examine the molecular mechanism of anti-inflammatory effect of phytochemicals from efficacious extracts.

### Funding

The financial support for this research was provided by the Collaborative Research and Development Grant program (CRDPJ 448052) of the Natural Sciences and Engineering Research Council (NSERC) of Canada and Island Abbey Foods, Charlottetown, PE, Canada. The funders had no role in study design, data collection and analysis, decision to publish, or preparation of the manuscript.

### Grant Disclosures

The following grant information was disclosed by the authors:
Collaborative Research and Development Grant program: CRDPJ 448052.
Natural Sciences and Engineering Research Council (NSERC) of Canada and Island Abbey Foods.

### Competing Interests

The authors declare there are no competing interests.

### Author Contributions

- Niluni M. Wijesundara performed the experiments, analyzed the data, wrote the paper, prepared figures and/or tables, reviewed drafts of the paper.
- Satvir Sekhon-Loodu performed the experiments, wrote the paper, reviewed drafts of the paper.
- HP Vasantha Rupasinghe conceived and designed the experiments, contributed reagents/materials/analysis tools, wrote the paper, reviewed drafts of the paper, principal investigator and supervisor.

### Data Availability

The raw data is included in the manuscript as the following format:

Table 2: mean and standard error of mean.

Table 3: mean and standard error of mean.

Figure 1: representative images for each treatment group.

Figure 2: representative images for each treatment group.

Figure 3: mean and standard error of mean.

Figure 4: mean and standard error of mean.

### Supplemental Information

Supplemental information for this article can be found online at http://dx.doi.org/10.7717/peerj.3469#supplemental-information.

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
