# Peer review of "Phytochemical-rich medicinal plant extracts suppress bacterial antigens-induced inflammation in human tonsil epithelial cells"

_PeerJ, doi:10.7717/peerj.3469_

## Round 0.1 · original submission · Major Revisions

The manuscript has been appreciated by the reviewers, however there is a concern on data validation by checking anti-inflammatory cytokines involvement. Also, the authors shoud pay attention to figures and tables as suggested.

Reviewer 1 ·

Basic reporting

the paper is interesting

Experimental design

The manuscript would benefit from inclusion of introducing/bridging sentences between the individual parts of the "Results" that explain the logical order and rationale for the experiments

Validity of the findings

In the Discussion, the Authors should highlight the possible clinical significance of their findings

Additional comments

The paper is interesting.
I have some question regarding this paper:
1. The extracts should contain a type of standardization (e.g. fingerprint).
2.The manuscript would benefit from inclusion of introducing/bridging sentences between the individual parts of the "Results" that explain the logical order and rationale for the experiments
3. In the Discussion, the Authors should highlight the possible clinical significance of their findings

Reviewer 2 ·

Basic reporting

no comments

Experimental design

no comments

Validity of the findings

no comments

Additional comments

In this study, the authors investigated the anti-inflammatory effects of extracts from 12 herbs on human tonsil epithelial cells (HTonEpiC).
The research is clear, well written and organized.
The introduction shows the background, methods of investigation and scope of the research.
Material and methods section is well organized but it would be appropriated to define the plant parts used for the extraction including all the detail of the extraction procedure and add the Table of reference where they were reported. In the calculation of the cell viability (%) (line 130), change “blank” with “Absorbance of the blank” at the first term of the equation. Add more details of the stop solution in line 155, 182. Line 184 put the acronym GCP-2 in bracket.
In the Results section, more details are needed. It would be appropriated to add differences of the extraction yield as AEs showed higher extraction yield than their respective EEs. Check the concentrations used for the cytotoxic effect, 0.5-100 µg/mL as reported in the results section (line 215) or 0.5, 1 and 5 µg/mL as reported in material and methods section (lines 121-122). Table 3 reports the cell viability data, and geranium L shows cytotoxic effects at ≥1µg/mL concentration; licorice root and sage reported higher cell viability >80% as other extracts (lines 215-218). Define if other extract concentrations were tested and data were not shown. Change p <.0.05 in p < 0.05 (line 237). Check lines 223-229 and lines 243-251, they report same results. In Table 3 change “theme” in “thyme”.
The Discussion section is well organized and discussed.

·

Basic reporting

Manuscript has been written well with professional English.
Sufficient background has been provided. There is need to change the figures and table.

Experimental design

No Comments

Validity of the findings

To validate the anti-inflammatory properties , anti-inflammatory cytokines should be studied.

Additional comments

In the current study authors have studied the anti-inflammatory properties of the aqueous and ethanol extracts of 12 different herbs. The manuscript has been well written with informative introduction and well-designed experiments along sufficient with methodology and discussion about the probable mechanism of anti-inflammatory properties of the herbs. Authors have detected some of the pro-inflammatory cytokines and chemokines, and also studied human beta defensing (hBD-2) which has microbicidal properties. The expression of proinflammatory cytokines has been reduced along with hBD-2 which is not good in terms of bacterial killing. Since LTA and PGN are TLR ligands, therefore the cytokines TNFa and IL-1beta should be studied. Along with these cytokines, other anti-inflammatory cytokines should be studied which would support the notion that these plant extracts have anti-inflammatory properties.
1) Table 3: Instead of table graph would be appropriate to see the effect clearly.
2) Figure 3 : It would be more informative to make a graph based on the absolute values of cytokines instead of percentage of cytokines.

---

## Round 0.2 · accepted · Accept

Dear Authors,

As requested by one of the reviewers, you are warmly invited to add the rationale for selecting the investigated specific cytokines for this cell model in the manuscript under Discussion section. Although not crucial, I believe the manuscript will get benefit with this addition and this can be done while in production.

Reviewer 2 ·

Basic reporting

'no comment'

Experimental design

'no comment'

Validity of the findings

'no comment'

Additional comments

'no comment'

·

Basic reporting

The language of the manuscript is correct.

Experimental design

No comments

Validity of the findings

no comments

Additional comments

Please add the rationale for selecting the investigated specific cytokines for this cell model in the manuscript under Discussion.
If authors think that Table 3 in the current format is suitable for the better readership, please keep it in the current form. The manuscript in the current format is suitable for publication.